# Chronic and Gradual-Onset Injuries and Conditions in the Sport of Surfing: A Systematic Review

**DOI:** 10.3390/sports9020023

**Published:** 2021-01-29

**Authors:** Samuel Hanchard, Ashley Duncan, James Furness, Vini Simas, Mike Climstein, Kevin Kemp-Smith

**Affiliations:** 1Water Based Research Unit, Faculty of Health Sciences, Bond University, Gold Coast, QLD 4207, Australia; ashley.duncan@student.bond.edu.au (A.D.); jfurness@bond.edu.au (J.F.); vsimas@bond.edu.au (V.S.); michael.climstein@scu.edu.au (M.C.); kkempsmi@bond.edu.au (K.K.-S.); 2Clinical Exercise Physiology, School of Health and Human Sciences, Southern Cross University, Bilinga, QLD 4225, Australia; 3Physical Activity, Lifestyle, Ageing and Wellbeing, Faculty Research Group, Faculty of Health Sciences, The University of Sydney, Lidcombe, NSW 2006, Australia

**Keywords:** epidemiology, musculoskeletal, non-musculoskeletal, injury, overuse, water sports, exostosis, aquatic

## Abstract

The majority of the previous literature investigating injuries in surfing have focused on acute or traumatic injuries. This systematic review appears to be the first to investigate the literature reporting on chronic and gradual-onset injuries and conditions in surfing populations. A search strategy was implemented on five databases in June 2020 to locate peer-reviewed epidemiological studies on musculoskeletal injuries or non-musculoskeletal conditions in surfing. A modified AXIS Critical Appraisal Tool was used to appraise all included texts. Extracted data included key information relevant to the epidemiology of the injuries and conditions. Twenty journal articles were included with the majority rated as good quality and a substantial agreement between raters (k = 0.724). Spine/back (29.3%), shoulder (22.9%), and head/face/neck (17.5%) were the most frequently reported locations of musculoskeletal injury, whilst the most common mechanism of injury was paddling (37.1%). Exostosis was the most frequently described injury or condition in surfing populations, with the most common grade of severity reported as mild obstruction. The key findings of injury type, location, severity, and mechanism can be used to develop relevant injury management and prevention programs for the surfing population, with an emphasis on chronic or gradual-onset spine/back and shoulder injuries, paddling technique, and education on the development and management of exostosis.

## 1. Introduction

Surfing continues to grow in popularity, with an estimated 35 million surfers worldwide and 2.7 million in Australia (10.6% of the estimated Australian population of 25.5 million) [1]. Approximately 7.7% of the world’s surfing population reside in Australia, which is interesting considering Australia only accounts for 0.33% of the world’s population [1]. Surfing’s worldwide popularity is expected to increase further in both recreational and competitive surfers following its inaugural showing at the 2021 Tokyo Olympics [2].

It is widely reported that surfers spend their time in four distinct categories: paddling, stationary, wave riding, and miscellaneous (e.g., wading, recovering after falling off the board, and duck diving), with both recreational and professional surfers spending approximately 50%, 40%, 3%, and 7% respectively in each category [3,4]. Professional and recreational surfers can spend up to four hours in the water during any one session [5,6], whereas surfing competitions usually consist of multiple 20 to 45-min heats [3]. The physical demands of competitive surfing has previously been investigated by Farley et al., [4] where competitive surfers were found to spend 58% of their paddling time at low speeds (1–4 km/h) followed by ≈30% at high-intensity speeds (4.1–8 km/h), with participants covering an average distance of 1605 m over two surfing events.

Long periods of prone paddling with isometric hyperextension of the cervical spine and repeated movements through maneuvers have been reportedly associated with overuse injuries within the sport of surfing [7,8]. Mendez-Villanueva and Bishop [3] discuss the importance of balanced muscular strength and flexibility of the shoulder, abdominals, back, and hamstrings, with any imbalances indicating a predisposing factor for injury. The introduction and increase in popularity of smaller surfboards, commonly known as shortboards, increases risk for injury, as a loss of buoyancy increases the difficulty of paddling, thus increasing stress on the shoulders whilst paddling [6]. The inclusion of aerial maneuvers in surfing competitions has been hypothesized as a reason for an increase in lower limb injuries, as the ability to perform and land these highly skilled maneuvers places increased ballistic stress on the lower limb joints and surrounding structures [7]. Acute injuries that are poorly managed, or conditions with residual symptoms, are believed to predispose athletes to re-injury. Loss of joint range of motion can cause further damage as a result of compensatory measures adopted by an individual following injury and muscle atrophy [9].

With an increase in studies reporting surfing injuries, there is confusion surrounding the definitions of injury causation and classification. Verhagen [10] categorized injury causation as either traumatic, being a single, specific, and identifiable event, or gradual onset, being a micro-trauma without a single or identifiable event. Similarly, Remnant et al. [11] discussed how gradual-onset injuries can either be chronic (i.e., long standing) or acute (i.e., recently made apparent). Conversely, previous literature has blended the causation and classification of time frames to define “chronic injury” as a condition that occurs over a period of time with no specific mechanism [12,13]. It is important when commenting on surfing injuries that both chronic and gradual onset definitions be considered. Very few studies have investigated both musculoskeletal injuries and non-musculoskeletal conditions, which may be due to difficulties associated with connecting conditions to a definitive cause or mechanism of injury. However, previous studies have suggested that consistent environmental exposure during surfing, such as prolonged time exposed to wind, sunlight, and water, can increase the likelihood of developing chronic, non-musculoskeletal conditions, such as exostosis or pterygium [12,14,15]. Exostosis is a condition defined by the presence of benign bony tumors in the external auditory canal, which are developed from the temporal bone [16,17,18]. Whilst the exact mechanism is still debated, individuals may experience hearing impairments due to water trapping and stenosis of the external auditory canal (EAC), otitis externa (inflammation of the ear canal), and otalgia (earache) [15]. Pterygium is a fibrovascular growth from the limbus to the cornea of the eye [19]. Pterygium rarely results in in blindness; however, a change in the curvature of the corneal surface can obstruct the field of vision [19]. As non-musculoskeletal conditions can be chronic or gradual onset in nature, it is important that these conditions are also taken into consideration with musculoskeletal injuries.

To the author’s knowledge, there is currently no systematic review that has investigated and collected data regarding both chronic and gradual-onset musculoskeletal injuries and non-musculoskeletal conditions in surfing. Coupled with the fact that most of the previous research has focused on acute injuries, the aim of this systematic review was to collate, critically appraise and summarize the available research investigating the epidemiology of chronic and gradual-onset injuries and conditions in the sport of surfing.

## 2. Materials and Methods

This study followed the methodology proposed in the Preferred Reporting Items for Systematic Reviews and Meta-Analysis (PRISMA) [20]. The protocol was registered with the Open Science Framework (OSF) in June 2020 (access via osf.io/gnhfu) [21].

This review aimed to collate articles that focused on the surfing population only and excluded other water sports, such as wakeboarding, waterpolo, and waterskiing. To be included, the articles were required to report on prevalence or incidence of chronic or gradual-onset injuries or conditions affecting the participants. Therefore, a cross-sectional study design was the most appropriate design to be included in this review, as the design can be used to investigate prevalence in population-based surveys. To create the search strategy, a layered approach was used. A limited, simple search was initially conducted to gather articles and key words relevant to the research question and topic. A collection of the initial articles sourced was collated, and the reference lists of all included studies were searched to identify any other articles to include. Then, the search was further refined using key words and relevant MeSH terms in the electronic databases PubMed, Embase, CINAHL, SPORTDiscus, and Google Scholar.

Our search strategy for the PubMed database is included below (Table 1), and minor alterations were made to allow for valid searches using Embase, CINAHL, and SPORTDiscus. The search strategy for Google Scholar was altered further to fit the search box (Table 1).

The Canadian Agency for Drugs and Technologies in Health (CADTH) gray literature searching tool was used to identify any non-indexed literature of relevance to this review. As per the CADTH, we incorporated the first 100 hits (i.e., first 10 pages of results) from Google Scholar into our study selection [22].

Studies were included or excluded according to the eligibility criteria outlined in Table 2.

A reference management software (EndNote, Version X9.3.3, Clarivate Analytics, Philadelphia, PA, USA) was used to import search results. Following the removal of duplicate articles, results were independently screened by title and abstract by two authors (A.D. and S.H.). A third reviewer (JF) was used when the two primary reviewers were unable to agree or unsure on the inclusion of the paper. For example, epidemiological articles that did not clearly define the nature of the injuries reported were discussed with the third reviewer due to the uncertainty surrounding their inclusion. As the reviewers could not determine whether the injuries were traumatic, acute, gradual onset, or chronic, the papers were excluded.

Studies eligible for a full-text review were re-assessed by two authors (A.D. and S.H.) independently. The results of the search strategy and process were included in the format of a PRISMA flow diagram.

A modified version of the AXIS Critical Appraisal Tool [23] was adapted from the tool utilized by McArthur et al. [24] and used to examine the quality of each of the included studies. The AXIS tool can be implemented to assist the inclusion of cross-sectional studies within systematic reviews [23]. Further modification of the tool was conducted to ensure all the questions were relevant to chronic and gradual-onset injuries and conditions. The exclusion of Question 12 from the tool by McArthur et al. [24] was justified, as the question was not applicable to the research question being investigated. This resulted in a total of 24 questions. Question 14 was modified with ‘2’ points being allocated for data collected objectively opposed to ‘1’ point for data collected via self-reporting. Self-reported data are susceptible to recall bias, which therefore may negatively affect the validity and credibility of the study [25]. Question 15 was modified, with the addition of an “if appropriate” option, to ensure non-musculoskeletal conditions were accounted for. The wording of Questions 17 and 23 was modified from the original AXIS Critical Appraisal Tool to ensure a numerical sum of all scores. For example, nil “concerns about non-response bias” or nil “funding sources or conflicts of interest” would result in ‘1’ point. The maximum attainable score was ‘25’, compared to ‘27’ in the tool modified by McArthur et al. [24] and ‘20’ in the original AXIS Critical Appraisal Tool for Cross-Sectional Studies [23]. A quality grade was applied as per the Kennelly Rating System and adapted in line with the research completed by McArthur and colleagues [24,26]. The raw AXIS scores were converted into percentages, with a score equal to or greater than 74% considered ‘good’, a score between 55% and 73.9% considered ‘fair’, and a score less than 54.9% considered ‘poor’. The modified AXIS Critical Appraisal Tool can be found in Appendix A.

The appraisal process was conducted by two reviewers (A.D. and S.H.) independently, with the interrater reliability of the two authors’ appraisal scores determined, in SPSS (Version 24.0, IBM SPSS Statistics for Windows, Version 24.0. IBM Corp., Armonk, NY, USA), using the Cohen’s Kappa Coefficient (k) [27].

Extraction of the data was conducted independently by two authors (A.D. and S.H.), which included all key information relevant to gradual-onset injuries and conditions. Examples included the following: author, year of publication, title, population number and demographics, location of injury, type of injury, mechanism of injury, severity of injury, geographical location of injury, study design, the level of evidence, as well as incidence and prevalence data. Then, the extracted information was collated into a data extraction table after an original pilot that was checked and approved by all authors.

To report on the information extracted, data were presented in tabular format for data collection method, data collection setting, geographical location, and population demographics. The absolute values were presented in tabular format for the total number of injuries, type of injuries, body region or location affected, mechanism of injury, and prevalence. For injury location and mechanism of injury, absolute frequencies were summed together and divided by the total number of injuries sustained to provide an overall frequency value. Due to the heterogeneity of grouping across the studies, specifically for body location and mechanism of injury, a simplification process was implemented to collate the data from all the included articles. This process was based upon the categorization utilized by McArthur and colleagues [24]. All injuries and conditions were sorted into nine categories for body location of injury, as shown in Appendix A, and ten categories for mechanism of injury, as shown in Appendix A.

## 3. Results

### 3.1. Data Search Results

The literature search process is presented using a PRISMA flow diagram (Figure 1), which illustrates the selection of articles, removal of duplicates, and final screening. A total of 1642 articles were found through the database search, which was inclusive of one hand-picked article that was not picked up in the database search and accessed through the author’s previous research. This was lowered to 1017 after duplicates were removed. The exclusion of 920 articles through title and abstract screening resulted in 97 articles that were eligible for full text screening. Seventy-seven articles were excluded due to reasons listed in Figure 1. A total of 20 studies were included for data extraction and synthesis.

### 3.2. Critical Appraisal Results

Out of the 20 articles included, eleven [11,14,15,16,19,28,29,30,31,32,33] were rated as ‘Good’ quality, six [12,17,34,35,36,37] were rated as ‘Fair’ quality, and three [5,6,18] were rated as ‘Poor’ quality, with a mean of 69.9% (SD ± 11.27%) and ranges between 45.8% and 84.0% (Table 3). The Cohen’s Kappa (k) yielded a ‘substantial agreement’ (k = 0.724) between raters for the critical appraisal [27]. Final results are presented in Table 3. All articles were rated as either Level II or Level III-2 on the National Health and Medical Research Council (NHMRC) levels of evidence [38].

### 3.3. Study Characteristics and Main Findings

All eight articles reporting on musculoskeletal injuries were cross-sectional studies published between 1983 and 2020 [5,6,11,12,28,29,32,37]. Five of the studies were conducted using researcher-administered questionnaires available at surfing competitions or within clinical settings [5,6,28,32,37]. The remaining three studies were conducted with online survey questionnaires [11,12,29].

Fifteen articles reported on non-musculoskeletal conditions [12,14,15,16,17,18,19,29,30,31,32,33,34,35,36]. One article focused solely on pterygium [19], eleven articles focused only on exostosis [14,15,16,17,18,30,31,33,34,35,36], while three articles looked at chronic health conditions, combining both musculoskeletal and non-musculoskeletal injuries and conditions [12,29,32]. Data focusing on exostosis were collected through researcher-administered questionnaires and otological examinations [14,15,16,17,18,30,33,34,35,36] or via self-reporting using a questionnaire or online survey [12,29,31,32]. Pterygium was recorded through similar objective methods including a researcher-administered questionnaire and penlight examination [19] and via self-reporting through a questionnaire [29,32]. Non-musculoskeletal conditions were reported between 1986 and 2020 with six articles collecting data at surfing events [14,16,18,30,33,34], four at popular beaches [19,32,35,36], two over the internet [29,31], one in a retail store [17], and one in a research clinic [15] (Table 4).

There were 4499 cases of musculoskeletal and non-musculoskeletal injuries and conditions documented in the data. Of these cases, 1941 were of non-musculoskeletal origin, and the remaining 2558 cases were musculoskeletal.

### 3.4. Type of Injury

Overuse musculoskeletal injuries were directly reported on three occasions [5,6,12], while chronic musculoskeletal injuries were directly reported on four occasions [28,29,32,37]. Gradual-onset injuries were reported in only one study [11]. These injuries were classified as either acute (n = 240 [<3 months]) or chronic (n = 310 [>3 months]), referring to the time taken to recover [11]. Of the eight articles reporting on musculoskeletal injuries, Furness et al. [29] was the only article to investigate the specific origin of injuries, with joint origin (43.5%) being the most common site, which was followed by muscular origin (23.6%). Similarly, Inada et al. [37] focused solely on injuries of joint origin. Chronic lower back pain was only reported within one study, by Bazanella et al. [28], and it was reported in 53% of the participants. However, the back was frequently recorded as an injured body region within other studies [11,12]. Chronic injuries were the most common injury type in three articles [28,29,37] (Table 5).

Non-musculoskeletal conditions were reported as conditions of the eye (e.g., pterygium) or ear (e.g., exostosis). Twelve of the non-musculoskeletal studies solely examined one non-musculoskeletal condition, with 100% of the data extracted originating from either exostoses or pterygium [14,15,16,17,18,19,30,31,33,34,35,36] (Table 5).

Musculoskeletal and non-musculoskeletal injuries or conditions were reported together in three studies, with non-musculoskeletal conditions accounting for 33% [12], 7.7% [29], and 51.4% [32] of reported injuries and conditions. A full breakdown of injury type, body region, mechanism and severity from each article can be seen in Appendix A whilst the most common injury type, body region and mechanism can be seen in Table 5.

### 3.5. Location of Injury

A total of 2115 musculoskeletal injuries were described, in conjunction with region of the body injured, within six of the eight articles reporting on musculoskeletal injuries [11,12,28,29,32,37]. The spine/back and the shoulder were consistently mentioned as the most common injury sites, representing 29.3% and 22.9% of the injuries recorded, respectively. This was followed by the head/face/neck (17.5%) and knee (10.4%). The full distribution of body region affected in musculoskeletal injuries can be seen in Figure 2.

The only non-musculoskeletal injuries and conditions reported within the literature were pterygium and exostoses, meaning that the body location or region was limited to eyes and ears, respectively. With the majority of the literature focusing on exostosis, the authors decided to exclude non-musculoskeletal conditions from this section in the results and to only focus on musculoskeletal injuries to gain a relevant representation of the body location of injury.

### 3.6. Mechanism of Injury

Five studies included in the analysis directly reported on the mechanism of injury, with a total of 2429 mechanisms collected across the data [5,6,11,12,29]. The most common mechanism reported throughout the literature was paddling (37.1%) followed by injuries associated with riding the wave (15.9%). Other mechanisms that were reported across the studies include maneuvers/aerials (11.8%), overuse of joint (11.7%), and unknown (10.1%). The full distribution of mechanism in musculoskeletal injuries can be seen in Figure 3.

### 3.7. Severity of Injury

Nine articles commented on the severity of exostosis with the majority of the articles using a scale noting the severity of ear canal stenosis. The majority of the participants were in groups depicting none or small amounts of ear stenosis, with all but one article reporting a decrease in exostosis cases with an increase in severity [15].

Five articles used a 4-point scale depicting stenosis of the ear canal [14,15,33,34,36]. Of the 1413 participants involved in these articles, 34.61% of participants presented with grade 0 (0% obstructed), 34.0% presented with grade 1 (1–33% obstructed), 17.7% presented with grade 2 (34–66% obstructed), and 13.7% presented with grade 3 (>66% obstructed) (Figure 4). Altuna et al. [35] commented on the amount of obstruction through a scale of <25% (n = 16), 25–50% (n = 19), 50–75% (n = 8), and >75% (n = 3) obstruction (Appendix A). Kroon et al. [16] used a 3-point scale with 62% of participants reporting no stenosis, 26% reporting mild stenosis, and 12% reporting moderate to severe stenosis (Appendix A). Umeda et al. [18] observed 60.7% of participants with normal to slight stenosis, 19.7% with unilateral severe stenosis, and 19.7% with bilateral severe stenosis. Only one study investigated the laterality of exostosis with 21.3% of participants suffering from bilateral exostosis compared to 7.6% participants with unilateral exostosis [31].

Only one of the three articles that reported on pterygium commented on the severity of the condition [19]. A 3-point grading scale was used, with 64.7% of cases graded as T1 (least severe), 17.6% graded as T2, and 17.6% graded as T3 (most severe). More than half of the cases (52.9%) were from the surfing enthusiast group, whilst 17.6% were from the recreational group, and 29.4% were from the occasional surfer group. A full breakdown of injury type, body region, mechanism and severity from each article can be seen in Appendix A

### 3.8. Geographical Location of Injury

Fourteen articles gathered data on exostoses in the sport of surfing [12,14,15,16,18,29,30,31,32,33,34,35,36,36]. All of them reported on the geographical location of the data collection, with the majority of the articles each collecting data within a single country. The only study to collect data from multiple different countries was by Nathanson et al. [12], whereby the data were gathered from Australia, England, New Zealand, and the United States. Seven countries were represented among the fourteen studies with four from Australia [12,15,29,32], three from England [12,34,36], one from Ireland [30], two from Japan [14,18], two from New Zealand [12,31], one from Spain [35], and four from the United States [12,16,17,33] (Figure 5).

## 4. Discussion

The aim of this systematic review was to collate, critically appraise, and summarize the available research investigating the epidemiology of chronic and gradual-onset injuries and conditions in the sport of surfing. The average methodological rating for the studies included in this review is 69.9%, which is of fair quality and needs to be considered when evaluating and applying the results [24,26].

### 4.1. Type of Injury

The reporting on the type of injury appears to follow a timeline. All three of the articles that commented on overuse injuries were published before 2002 [5,6,12], all four of the articles that commented on chronic injuries were published between 2004 and 2018 [28,29,32,37], and the article by Remnant et al. [11], which focuses on gradual-onset injuries, was published in 2020. The change in terminology could be associated with the recent understanding that the term “chronic” does not take into consideration the onset of injuries [11]. Previous reviews, which have documented the injury types in surfing, have focused on anatomical structures (e.g., skin, soft tissue, and bone) [24]. While this may be suitable for the reporting of acute injuries, this review found inconsistencies within the literature that made the confirming the onset of the documented chronic or gradual-onset injury impossible. For example, Lowden et al. [5] documented soft tissue injuries (e.g., sprains and strains) at particular joints that may be of gradual onset or chronic nature; however, this was not specified.

The majority of studies included were focused on one injury type: exostosis in the ear. As these articles are focusing on only one surfing condition, we can see that exostosis is prevalent in the surfing population; however, from the cohort studies, we can see other medical conditions and musculoskeletal injuries are also highly prevalent in the surfing community.

Other conditions that were mentioned included sinusitis, otitis, and cellulitis [12,29,32]. These conditions represented a small portion of the non-musculoskeletal data and, unlike exostosis and pterygium, to the best of our knowledge, there are no articles looking individually into the epidemiology of sinusitis, otitis, and cellulitis in a surfing population. Due to the limited amount of data and lack of continuity across the reporting of the conditions, they were not considered a main finding in this review.

### 4.2. Location of Injury

The categorization of body location varied across the included literature, with one study by Furness et al. [29], dividing the results for body location into 11 different categories, while other studies focused on six categories or less [11,12,28,32,37]. A broad set of categories, as seen in Appendix A, were generated to achieve homogeneity when combining the data for analysis.

The higher percentage of chronic and gradual-onset injuries to the back and shoulder may be associated with the large proportion of time spent in a paddling position. This position consists of repetitive arm strokes that may predispose the surfer to overuse injuries, as reported in the study by Mendez-Villaneuva and Bishop [3]. Paddling also exposes the surfer to prolonged periods in cervical hyperextension, which may explain why head, neck, and face injuries made up the third largest percentage of all chronic or gradual-onset injuries reported.

To the author’s knowledge, there are currently no other systematic reviews that have collated information on chronic or gradual-onset musculoskeletal and non-musculoskeletal injuries and conditions in surfing; however, we were able to compare and contrast reviews focusing on acute injuries. In the studies conducted by McArthur et al. [24] and Nathanson et al. [39], the face, head, and neck region was the most commonly injured, which was followed closely by the lower limbs. It is important to understand that this difference in location of injury, between acute and chronic or gradual-onset injuries and conditions, may indicate a predisposition of certain body regions to traumatic or, conversely, to atraumatic or overuse injuries.

### 4.3. Mechanism of Injury

There were five articles [5,6,11,12,29] that described the mechanism of chronic or gradual-onset injuries. Although there is a paucity of data surrounding the mechanism of injury, previous systematic reviews focusing on acute injuries in surfing have discussed as little as two articles reporting on the mechanism, which is due to the difficulty in determining the exact cause [24,39]. As per the body location of injury, all reported mechanisms were divided into ten categories as shown in Appendix A.

The greater percentage of chronic or gradual-onset injuries caused by paddling could be due to a multitude of factors. Both recreational and competitive surfers spend approximately 50% of their time paddling during a surf [3,4]. This large amount of time spent paddling may contribute to the higher percentage of injuries that occur due to this phase of surfing. Chronic and gradual-onset injuries associated with the rotator cuff (e.g., tendinitis) may also be more prevalent in the surfing population due to the repetitive overhead movements of the paddling arm [40].

Previously conducted reviews focusing on acute surfing injuries were also limited in the amount of articles reporting on the mechanism of injury [24,39]. In the study by McArthur et al. [24], the most common mechanism was collision between the surfer and surfboard, which was followed by approaching a wave or performing a maneuver. Although it is not always the case, acute injuries are usually traumatic in nature, and the surfer can identify an exact time, place, and action where the injury occurred. By comparison, chronic and gradual-onset injuries usually occur gradually over time, and a surfer may not always be able to pinpoint an exact activity where the injury occurred. This is an important consideration when reviewing the scarcity of the available data on the mechanism of chronic and gradual-onset injuries.

As exostosis is not fully understood, it was hard to comment on the mechanism of the conditions. The studies used in this review showed limited knowledge toward the onset of the conditions with newer studies proposing that the mechanism of onset is from the exposure of cold water and wind, resulting in a protractive mechanism of the tympanic membrane.

### 4.4. Severity of Injury

Severity was only reported in the literature when focusing on non-musculoskeletal conditions. The data collection varied due to inconsistencies across the grading scales used to measure the level of severity. The included articles, focusing on exostosis, used five different grading scales, which created complications when attempting to combine the data for further analysis. The majority of the literature grading exostosis to the patency of the ear canal as either no obstruction or 100% patent (normal/grade 0), <33% obstructed or >66% patent (mild/grade 1), ≈33–66% obstructed or ≈65–33% patent (moderate/grade 2), and >66% obstructed or <33% patent (severe/grade 3) [14,15,33,34,36]. A similar grading system was used by Kroon et al. [16], where patency was grouped into normal (100% patent), mild (99–66% patent), and moderate to severe (<66% patent) and Altuna et al. [35], who classified exostoses cases as <25% obstruction, 25–50% obstruction, 50–75% obstruction, and >75% obstruction. Deleyiannis et al. [17] only reported on median severity scores based on experience of surfing, with greater obstruction for more experienced surfers. Conversely, Umeda et al. [18] clustered exostoses cases based on a 0–10 scale of obstruction (e.g., 10 for 0% obstruction, 4 for 60% obstruction, and 0 for 100% obstruction) with normal to slight stenosis (unilaterally) graded as 10-6, severe stenosis (unilaterally) graded as 5-0, and severe stenosis (bilaterally) graded as 5-0. The study by Lennon et al. [30] was the only study that collected data using otological examination that did not include a severity scale; similarly, Nathanson et al. [12], Furness et al. [29], and Taylor et al. [32] only conducted surveys and did not include severity.

Whilst severity is frequently reported in the literature, the development of exostosis is still debated [31]. However, there are multiple risk factors that have been associated with the development and progression of exostosis. The number of years an individual has surfed has been linked to the development of exostosis, with more experienced surfers and water sport participants exhibiting increased incidence and severity [16,17,41]. Similarly, multiple studies have associated cold water immersion [16,42,43] and the cooling effect from wind [31,44,45] with the development of exostosis. The articles included in this review, reporting on the severity of exostosis, collected data from various warm and cold-water climates on a global scale. As there was no large variation in the data relating to the severity, this may contribute to previous findings that exostosis may not be caused by the temperature of the water, but rather by other external, environmental factors, such as wind [15].

### 4.5. Strengths, Limitations and Future Research

To the best of our knowledge, this was the first review that examined chronic and gradual-onset injuries and conditions in surfing. The review was strengthened by the rigorous guidelines set out by the PRISMA protocol [20]. A thorough search strategy was implemented on multiple, prominent databases, and the included texts were screened and appraised independently by two reviewers using an appraisal tool relevant to the cross-sectional studies included.

This review was limited by the lack of objectively collected musculoskeletal injury data. Consequently, there is a potential of recall bias as a result of the data being collected from surveys [46]. As noted by another surfing epidemiological review, there are risks of having larger numbers of injured compared to uninjured surfers replying to online surveys, distorting the injury incidence and prevalence [24]. It may also inhibit the ability to pool the prevalence of each grade of exostosis severity due to the abundance of grading systems used.

Another limitation involved the lack of good quality articles included, with only 11 articles exhibiting a ‘Good’ rating. This is a critical finding, as it demonstrates the importance of assessing articles prior to their use and may illustrate a need for further control in the methods of each of the studies.

Spinal myelopathy is usually referred to as a progressive condition to the spine and may usually be classified as a chronic or gradual-onset injury [47,48]. Surfer’s myelopathy is a similar condition that has been documented throughout the literature as a non-traumatic, acute ischemic injury of the spinal cord, which predominantly occurs in novice surfers [49,50,51]. Due to the ambiguity of the classification of the condition, usually in a relatively inexperienced population of surfers, the research team decided to exclude myelopathy from the review during the full text eligibility assessment under the classification of “unable to distinguish chronic from acute injury”. Future research should aim to investigate the pathophysiology and epidemiology of surfer’s myelopathy to avoid further confusion in relation to the classification of injury and to explore whether the condition significantly contributes to injury burden within a surfing population.

Future research, focusing on investigating the epidemiology of chronic and gradual-onset injuries in surfing, must ensure that all data can be properly categorized as per the current injury classifications. This will allow for more accurate interpretation of the results and clear distinction between acute injuries and chronic or gradual-onset injuries and conditions when documenting data in research and also in the community. Additionally, using epidemiological data for the prevention and management of injuries and conditions may be helpful to ensure the safety of all surfing participants. This may involve investigation into the biomechanics of paddling and the underlying causes that may contribute to the most common chronic and gradual-onset injuries, involving the spine, back, and shoulder. As the most common chronic non-musculoskeletal condition, awareness and education of exostosis in the surfing community should be implemented to avoid secondary injuries relating to the resulting hearing loss and balance deficits associated with the condition.

## 5. Conclusions

The most common chronic and gradual-onset musculoskeletal injuries occur to the spine/back and shoulder regions and are commonly caused by paddling, which consists of repetitive arm movements and excessive extension of the back. Exostosis and pterygium are both non-musculoskeletal conditions prevalent in surfing populations; however, the most commonly reported condition associated with surfing was exostosis. Although grade 1 was the most frequently reported level of severity for exostosis, it is important to consider that minor obstruction of the ear canal may cause hearing loss and balance deficits, which can severely hinder surfing performance. The exact mechanism or etiology of both exostosis and pterygium is still debated in the literature.

The results presented in this review confirm the need for further research into prevention and management strategies, for chronic and gradual-onset injuries and conditions in the sport of surfing, due to the established injury prevalence and burden in the surfing population. Future research should focus on the development of injury prevention initiatives related to chronic and gradual-onset spine, back, or shoulder injuries, the effect of paddling technique or biomechanics on the development of chronic and gradual-onset injuries, and the establishment of education initiatives to raise awareness surrounding common non-musculoskeletal conditions associated with surfing, such as exostosis or pterygium.

## Figures and Tables

**Figure 1 sports-09-00023-f001:**
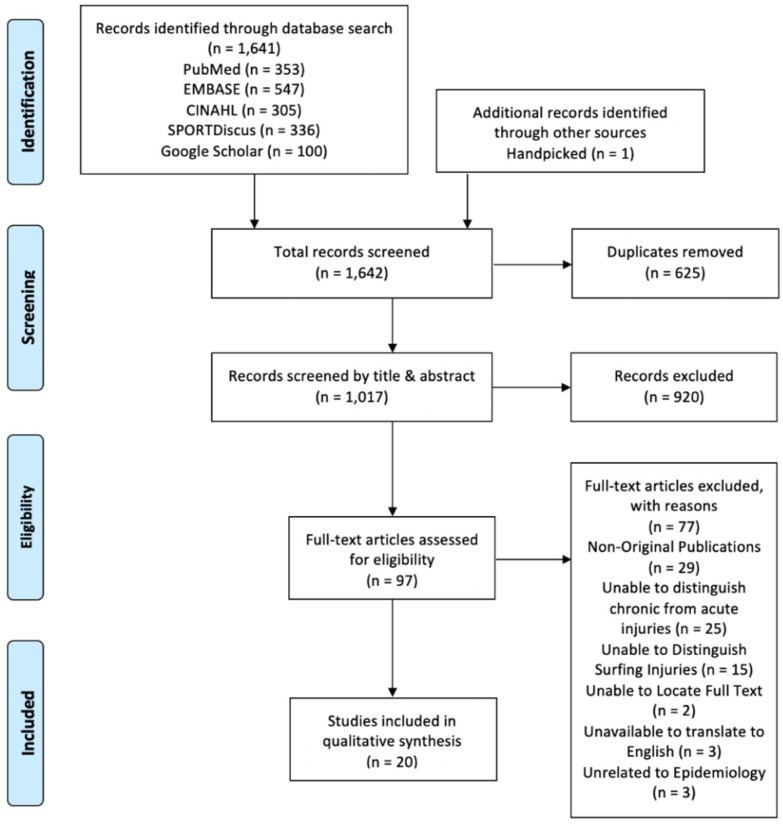
Preferred Reporting Items for Systematic Reviews and Meta-Analysis (PRISMA) flow diagram presenting results for literature search, screening and eligible studies.

**Figure 2 sports-09-00023-f002:**
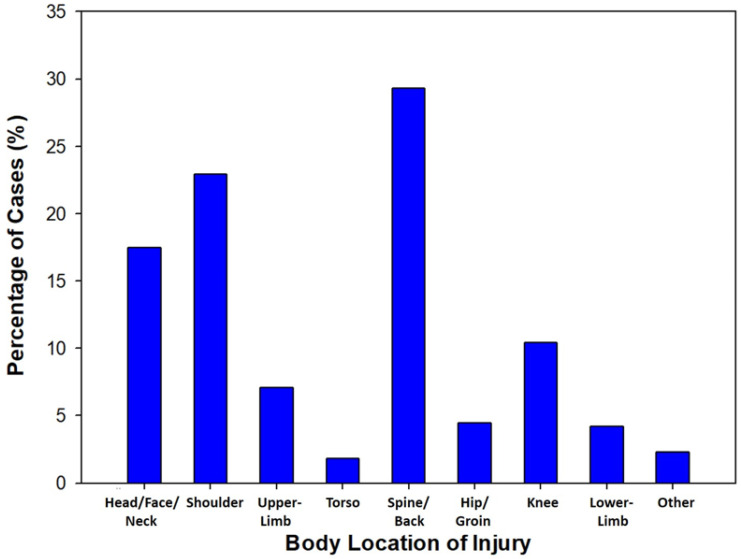
Column graph for location of injury (musculoskeletal only). six studies included [11,12,28,29,32,37].

**Figure 3 sports-09-00023-f003:**
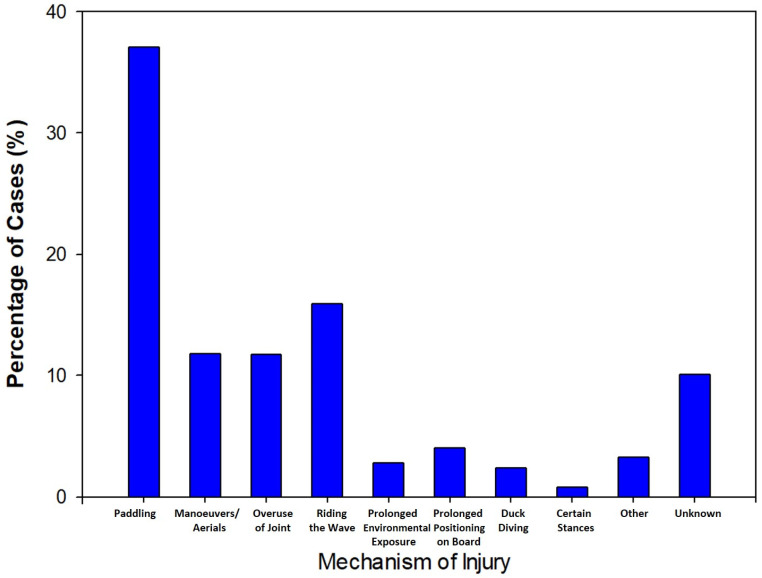
Column graph for mechanism of injury (musculoskeletal only). Five studies included [5,6,11,12,29].

**Figure 4 sports-09-00023-f004:**
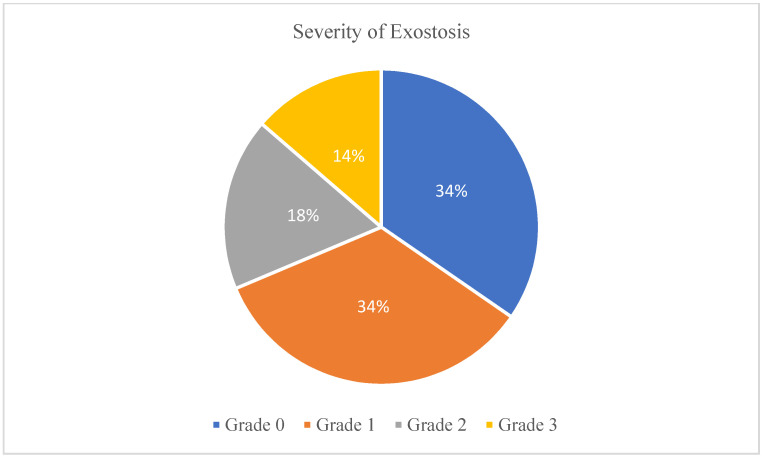
Pie chart of the percentage of cases of each severity grade. Five articles included [14,15,33,34,36]

**Figure 5 sports-09-00023-f005:**
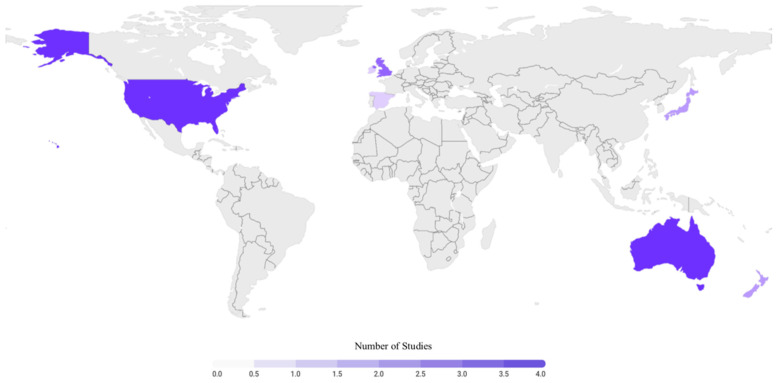
Global representation of included articles regarding exostoses. Fourteen articles included [12,14,15,16,17,18,29,30,31,32,33,34,35,36].

**Table 1 sports-09-00023-t001:** Search strategies for databases.

PubMed	Google Scholar Search Strategy
(**surfer*** OR **surfing** OR **surfboard** * OR “**surfboard riding**”) AND (**injury** OR **injuries** OR **disability** OR **disabilities** OR **illness** OR **non-musculoskeletal** OR “**non musculoskeletal**” OR **chronic** OR “**non acute**” OR **non-acute** OR **gradual** OR “**gradual onset**” OR **gradual-onset** OR **overuse** OR **atraumatic** OR “**Wounds and Injuries**[Mesh]” OR **exostosis** OR **exostoses** OR **pterygium** OR **myelopathy**)	(**surfer*** OR “**surfboard riding**”) AND (**injury** OR **injuries** OR **disability** OR **disabilities** OR **illness** OR **non-musculoskeletal** OR **chronic** OR **non-acute** OR **gradual** OR **onset** OR **overuse** OR **atraumatic** OR **exostosis** OR **exostoses** OR **pterygium** OR **myelopathy**)

* Alternate word endings have been automatically added into search strategy.

**Table 2 sports-09-00023-t002:** Eligibility Criteria (inclusion and exclusion criteria) with examples.

**Inclusion Criteria**	**Examples**
Original peer-reviewed publications Published studies documenting incidence or prevalence of chronic or gradual-onset musculoskeletal and non-musculoskeletal injuries/conditions	Journal articles or doctoral theses Chronic or gradual-onset injuries or conditions
**Exclusion Criteria**	**Examples**
Full text not available Full text not translatable to English Case reports or series Periodicals Data from surfing injuries cannot be interpreted independently from other water sport injury data Studies documenting data related to treatment or prevention of injuries or conditions only	Abstract only Mandarin, Japanese, French, etc. Studies documenting a case report of an individual Magazines or letters Surfing injuries mixed with bodyboarding, surf lifesaving, and bodysurfing Non-epidemiological studies

**Table 3 sports-09-00023-t003:** Study design, modified AXIS scores, allocated quality rating, and level of evidence for included studies.

Author (Year)	Title	Study Design	Modified Axis Final Score	Percentage (%)	Quality Rating	Level of Evidence (NHMRC)
Alexander et al. (2015) [34]	The effects of surfing behavior on the development of external auditory canal exostosis	Cross-sectional	17/24	70.8%	Fair	II
Altuna et al. (2004) [35]	Prevalence of exostosis among surfers of the Guipuzcoan Coast	Cross-sectional	16/24	66.7%	Fair	II
Attlmayr and Smith (2015) [36]	Prevalence of ‘surfer’s ear’ in Cornish surfers	Cross-sectional	15/24	62.5%	Fair	II
Bazanella et al. (2016) [28]	Association between low back pain and functional/kinetic aspects of surfers: disability, function, flexibility, range of motion, and angle of the thoracic and lumbar spine	Cross-sectional	19/25	76.0%	Good	III-2
Deleyiannis et al. (1996) [17]	Exostoses of the external auditory canal in Oregon surfers	Cross-sectional	14/24	58.3%	Fair	II
Furness et al. (2014) [29]	Retrospective Analysis of Chronic Injuries in Recreational and Competitive Surfers: Injury, Location, Type and Mechanism	Cross-sectional	21/25	84.0%	Good	III-2
Inada et al. (2018) [37]	Acute injuries and chronic disorders in competitive surfing: From the survey of professional surfers in Japan	Cross-sectional	15/25	60.0%	Fair	III-2
Kroon et al. (2002) [16]	Surfer’s ear: external auditory exostoses are more prevalent in cold water surfers	Cross-sectional	18/24	75.0%	Good	II
Lennon et al. (2016) [30]	Auditory canal exostoses in Irish surfers	Cross-sectional	19/24	79.2%	Good	II
Lin et al. (2016) [19]	Prevalence of pterygia in Hawaii: Examining cumulative surfing hours as a risk factor	Cross-sectional	19/24	79.2%	Good	II
Lowdon et al. (1983) [5]	Surfboard-riding injuries	Cross-sectional	13/25	52.0%	Poor	III-2
Lowdon et al. (1987) [6]	Injuries to international competitive surfboard riders	Cross-sectional	13/25	52.0%	Poor	III-2
Nakanishi et al. (2011) [14]	Incidence of external auditory canal exostoses in competitive surfers in Japan	Cross-sectional	19/24	79.2%	Good	II
Nathanson et al. (2002) [12]	Surfing injuries	Cross-sectional	17/25	68.0%	Fair	III-2
Remnant et al. (2020) [11]	Gradual-onset surfing-related injuries in New Zealand: A cross-sectional study	Cross-sectional	19/25	76.0%	Good	III-2
Simas et al. (2020) [15]	The prevalence and severity of external auditory exostosis in young to quadragenarian-aged warm-water surfers: A preliminary study	Cross-sectional	20/24	83.3%	Good	II
Simas et al. (2019) [31]	Lifetime prevalence of exostoses in New Zealand surfers	Cross-sectional	19/24	79.2%	Good	III-2
Taylor et al. (2004) [32]	Acute injury and chronic disability resulting from surfboard riding	Cross-sectional	19/25	76.0%	Good	III-2
Umeda et al. (1989) [18]	Surfer’s ear in Japan	Cross-sectional	11/24	45.8%	Poor	II
Wong et al. (1999) [33]	Prevalence of external auditory canal exostoses in surfers	Cross-sectional	18/24	75.0%	Good	II
				k = 1.000	Mean = 69.9% (SD ± 11.27)	

Study quality identified with color coding where green indicates ‘Good’ quality, blue indicates ‘Fair’ quality, and yellow indicates ‘Poor’ quality.

**Table 4 sports-09-00023-t004:** Method and setting of data collection and population demographics for included studies.

Author (Year)	Data Collection Method	Data Collection Setting	Geographical Location	Population Demographics			
				Number of Participants (n)	Mean Age (yrs)	Sex	
						M	F
Alexander et al. (2015) [34]	Researcher administered questionnaire and otological examination	Three surfing events in South West of England, Plymouth University, and Cornwall College	England	209	M = 26 F = 23	173	36
Altuna et al. (2004) [35]	Researcher administered questionnaire and otological examination	Surfers on the Guipuzcoan coast	Spain	41	29	39	2
Attlmayr and Smith et al. (2015) [36]	Researcher administered questionnaire and otological examination	Over two-week period (“two good swells were hitting the north Cornish coast”) Five beaches including South Fistral, St Agnes, Tolcarne, Perranporth, Porthtown	United Kingdom	105	30.08	92	13
Bazanella et al. (2016) [28]	Researcher administered questionnaire (with Nordic Musculoskeletal Questionnaire (NMQ) and QUEBEC Back Pain Disability Scale Questionnaire)	Professional, amateur, or recreational surfers from coast of Paraná (surfing for >6 months)	Brazil	66	25.7	50	16
Deleyiannis et al. (1996) [17]	Researcher administered questionnaire and otological examination	Two surf shops in northern Oregon coastal area (advertised on flyers and local daily newspaper)	USA	21	31.3	21	-
Furness et al. (2014) [29]	Researcher administered questionnaire	Free access online survey (using SurveyMonkey)	Australia	1348	35.8	1231	117
Inada et al. (2018) [37]	Researcher administered questionnaire and medical records	Records from Japan Pro Surfing Tour between 2009 and 2016 and outpatient clinic dedicated for professional surfing competitors	Japan	62	-	-	-
Kroon et al. (2002) [16]	Researcher administered questionnaire and otological examination	Day surfing competition at 2000 East Coast Surfing Championship	USA	202	19	186	16
Lennon et al. (2016) [30]	Researcher administered questionnaire and otological examination	Two regional surf competitions	Ireland	119	-	102	17
Lin et al. (2016) [19]	Research administered questionnaire and examination	Beach attendees at two beaches (Waimea Bay and Ehukai Beach Park) from northern shorelines	Hawaii	81	-	57	24
Lowdon et al. (1983) [5]	Researcher administered questionnaire	Reply-paid questionnaire sent to Victorian branch of Australian Surfriders Association	Australia	346	21.8	328	18
Lowdon et al. (1987) [6]	Researcher administered questionnaire	1982 Bells Beach international surfing Competition, Victoria	Australia	86	22.4	79	7
Nakanishi et al. (2011) [14]	Researcher administered questionnaire and otological examination	Five surfing competitions in Miyazaki, Japan	Japan	373	33.5	309	64
Nathanson et al. (2002) [12]	Researcher administered questionnaire	Free access online survey	USA, Australia, England, New Zealand	1348	28.6	1213	135
Remnant et al. (2020) [11]	Researcher administered questionnaire	Free access online survey	New Zealand	1437	34.6	1178	259
Simas et al.(2020) [15]	Researcher administered questionnaire and otological examination	Water-Based Research Unit Clinic	Australia	23	35.4	19	4
Simas et al. (2019) [31]	Researcher administered questionnaire	Online survey	New Zealand	1376	34.9	1123	253
Taylor et al. (2004) [32]	Researcher administered questionnaire	Eight Victorian beaches	Australia	646	27	583	63
Umeda et al. (1989) [18]	Researcher administered questionnaire and otological examination	Surfing competition	Japan	94	-	-	-
Wong et al. (1999) [33]	Researcher administered questionnaire and otological examination	Surfing competition	USA	300 (600 ears)	-	-	-

Study quality identified with color coding where green indicates ‘Good’ quality, blue indicates ‘Fair’ quality and yellow indicates ‘Poor’ quality.

**Table 5 sports-09-00023-t005:** Most common injury type, body region, and mechanism by article.

Author (Year)	Total Injury	Type of Injury (% of Total)	Body Region (% of Total)	Mechanism (% of Total)
Alexander et al. (2015) [34]	112	Exostosis (100%)	Ear (100%)	-
Altuna et al. (2004) [35]	25	Exostosis (100%)	Ear (100%)	-
Attlmayr and Smith (2015) [36]	134 ears	Exostosis (100%)	Ear (100%)	-
Deleyiannis et al. (1996) [17]	21	Exostosis (100%)	Ear (100%)	-
Kroon et al. (2002) [16]	76	Exostosis (100%)	Ear (100%)	-
Lennon et al. (2016) [30]	79	Exostosis (100%)	Ear (100%)	-
Nakanishi et al. (2011) [14]	223	Exostosis (100%)	Ear (100%)	-
Simas et al. (2020) [15]	16	Exostosis (100%)	Ear (100%)	-
Simas et al. (2019) [31]	397	Exostosis (100%)	Ear (100%)	-
* Taylor et al. (2004) [32]	146	Exostosis (45.9%)	Ear (45.9%)	-
Umeda et al. (1989) [18]	73 ears	Exostosis (100%)	Ear (100%)	-
Wong et al. (1999) [33]	441 ears	Exostosis (100%)	Ear (100%)	-
Lin et al. (2016) [19]	17	Pterygium (100%)	Eye (100%)	-
Bazanella et al. (2016) [28]	45	Chronic low back pain (100%)	Lumbar spine (100%)	Surfing (53%)
* Furness et al. (2014) [29]	1068	Chronic injuries of joint origin (43.5%)	Shoulder (23%)	Prolonged paddling (21.1%)
Inada et al. (2018) [37]	62	Chronic injuries or conditions of joint origin (100%)	Lower Back (31%)	-
Lowdon et al. (1983) [5]	337	-	-	Stress from maneuver (39%)
Lowdon et al. (1987) [6]	187	-	-	Stress from maneuver (42%%)
* Nathanson et al. (2002) [12]	477	Overuse syndromes (62%)	Shoulder (18%)	Overuse of shoulder (18%)
Remnant et al. (2020) [11]	550	Gradual-onset injuries (100%)	Shoulder (27%)	Prolonged paddling (28%)

Key: * Musculoskeletal and non-musculoskeletal included in study. Study quality identified with color coding where green indicates ‘Good’ quality, blue indicates ‘Fair’ quality and yellow indicates ‘Poor’ quality.

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
