# Peer review of "Chronic and Gradual-Onset Injuries and Conditions in the Sport of Surfing: A Systematic Review"

_sports, 2021, doi:10.3390/sports9020023_

Round 1
Reviewer 1 Report
Please see attached document

Reviewer 2 Report
This is an interesting epidemiological literature search on the onset of injuries in surfing (chronic and gradual). According to the authors, it represents the first review on this topic. The main contribution of this article is that with the added knowledge on onset of injury, management and prevention strategies can be planned in this sporting activity. The paper is excellently written: clear, concise and systematic. The methods are well described so that the reader can repeat the study. The results provided re well discussed and put into perspective with previous research. The conclusions are reflecting what was discussed and gives directions for further research.
Accept as is!
Reviewer 3 Report
- Keywords, please use different terms than title.
- The introduction section is clear. However, the aim of the study needs to be clarified because the need of this study should not be focused on the non existence of previous systematic reviews. On the contrary, it would be the need for a better understanding of injuries in surfing. Please rewrite the aim in the same way than discussion section.
- Methods, as the authors used PRISMA protocol, please clarify the PICOS procedure.
- Data bases used, please explain why Web of Science and Scopus were omitted in this review. Both are the most relevant databases in the world. Please include the same search in both databases in order to add any missing study.
- L130, please cite correctly the statistical software: https://www.ibm.com/support/pages/how-cite-ibm-spss-statistics-or-earlier-versions-spss
- L131, L165, please write Cohen’s instead of Cohens
- Please revise text in L140-150 as you used different size letter.
- Limitations, please raise the importance of assessing articles. In the current study only 11 were evaluated as good. Then, it means that research designs in this topic need to control for the limitations of procedures and methods of fair and poor studies of injuries in surfing.
Round 2
Reviewer 1 Report
Nice work in revising your manuscript. It is much improved and much more clear. I still have one concern that need to be addressed:
Lines 34 and 35: there is still no comparison of percentages in Australia vs the rest of the world. Its remarkable that 10% of Australians surf but compare that to the rest of the world to make your point. Either by: saying what percentage of the rest of the world population surfs OR as I mentioned previously state that Australia only contains .33% of the world population but 7.7% of the population of surfers.
Outside of that you have responded to all my my previous comments satisfactorily.
